# Development of the Psychological Capital Scale for Male Nursing Students in Taiwan and Testing Its Measurement Invariance between Genders

**DOI:** 10.3390/ijerph19063620

**Published:** 2022-03-18

**Authors:** Jiunnhorng Lou, Renhau Li, Shuling Chen

**Affiliations:** 1Department of Nursing, Hsin Sheng College of Medical Care and Management, Taoyuan 325004, Taiwan; stujhl@gmail.com; 2Department of Psychology, Chung-Shan Medical University, Taichung 40201, Taiwan; davidrhlee@yahoo.com.tw; 3Department of Nursing, Hungkuang University, Taichung 433304, Taiwan

**Keywords:** male nursing students, psychological capital, hope, optimism, resiliency, self-efficacy, measurement invariance, brief scale

## Abstract

The aims of this study were to develop a psychological capital (PsyCap) scale for male nursing students and to compare the scores with those of female nursing students. Few past studies have focused on male nursing students to understand their PsyCap relative to female nursing students. We recruited 384 male nursing students in Taiwan to construct the PsyCap Scale with 16 items and four factors based on the relevant literature: hope, optimism, resiliency, and self-efficacy. The scale showed good model fit in confirmatory factor analysis with factor loadings from 0.62 to 0.78. Cronbach’s alpha coefficients ranged from 0.75 to 0.83 for the four subscales and 0.91 for the total scale. We also conducted measurement invariance tests with data from 402 female nursing student volunteers in Taiwan. The invariance of factor loadings and intercepts of the established scale (i.e., with the same unit and origin between genders) indicated that the male nursing students had higher PsyCap in optimism and resiliency than the females. We developed a 16-item-scale to make administration rapid and convenient and applied advanced statistical methods for reliable and valid comparisons between sexes. The results may help the government to create education programmes or policies supporting male nursing students.

## 1. Introduction

Nursing has long been considered a profession predominately for women, and ‘nursing knowledge’ is equal to ‘women’s knowledge’ from the perspective of traditional occupational and social culture. Therefore, the public image of nurses is tightly connected to that of good women and good mothers. As there are fewer studies, it is not known whether this perception of nursing decreases the psychological capital (PsyCap) of male nursing students and it therefore deserves further exploration.

PsyCap is an individual’s positive psychological resource [1] that reflects a positive psychological state in their personal growth and development. It enables self-efficacy and characterises an individual’s advantages, thus allowing them to realise their valuable potential [2]. Additionally, PsyCap encompasses invisible capital and positive energy, and it focuses on personal positive psychological development and strives to present personal good or positive facets, which enable one to overcome frustration and face hard times in life with a positive attitude [3]. Da et al. [4] argued that people with abounding PsyCap expect to have positive experiences at work, believe they are capable of achieving success, are unlikely to be influenced by setbacks, and are willing to help others. If students persist in their self-efficacy and resiliency, they are likely to earn good grades and have success in academic achievement [5,6]. Therefore, PsyCap emphasises positive psychological development, thus helping accumulate psychological resources [7]. PsyCap includes hope, optimism, resiliency, and self-efficacy [8,9]. Luthans, Youssef, and Avolio [10] described the four constructs as follows: hope—an approach to acquiring success with volition and effort; optimism—positive expectations about the future and attributions to the present; resiliency—coping well with adversity and negativity, and recovering quickly from setbacks to achieve success; and self-efficacy—the confidence to obtain success by making optimum efforts when facing challenging tasks. An et al. [11] thought that when someone has high PsyCap, they can offer more resources to organisational operations, maintain better relationships with colleagues, have optimistic perceptions and positive self-affirmation, and achieve positive performance.

PsyCap is a kind of psychological state with progressive development, thus significantly affecting nursing students’ professional identification and nursing careers [12,13]. It can promote personal positive psychological development, help nursing students reach their present and future goals, and overcome obstacles to success [14]. Hence, PsyCap is viewed as an index to assess factors involving one’s positive psychological development, such as success, learning satisfaction, well-being reinforcement, and achieving desirable goals [15]. Students with lower PsyCap had higher dropout rates [16]; however, nursing students with higher PsyCap would take different viewpoints towards learning stress and perform better in learning adjustment [17]. Eun and Mi-Ra [18] pointed out that PsyCap has a significant positive correlation with academic achievement. Terry et al. [19] also found that PsyCap helped nursing students in their study process by reinforcing their ability to overcome difficulties. However, Manoochehri et al. [20] found that gender has different relationships with PsyCap and spirituality. Pan et al. [21] found that PsyCap helped male nurses in the practice environment to develop their professional ability fully and increase their intent to engage in work. Some research has also found that male nursing students want people to refer to them as nurses rather than male nurses [22]. These studies imply that more research should be conducted to understand male nursing students in terms of variables, such as PsyCap, in comparison to female nurses.

In summary, PsyCap also has a positive influence on male nursing students’ development of their nursing careers, professional performance, and role orientation. However, only a few studies have investigated the PsyCap of male nursing students relative to that of female nursing students; thus, more efficient education programmes for them cannot be developed. Whether the advantages of PsyCap for female nursing students are similar to those for male nursing students would need more research. In the present study, the PsyCap Scale for male nursing students was developed, and its differences relative to female nursing students were tested using advanced statistical methods.

## 2. Materials and Methods

The study was of an explanatory research design type, which employed a pragmatic methodology in terms of approach. It included a qualitative approach to literature review for writing a draft of the PsyCap Scale and for its scrutiny by experts, and a quantitative approach to data collection and statistical analysis using a cross-sectional survey method. The questionnaire included gender, age, general learning experiences, and a draft of the PsyCap Scale. The study was approved by the Institutional Review Board (IRB No. 202110-E101).

### 2.1. Procedures

In accordance with the definitions of PsyCap, a draft with 31 items was first developed by the authors, including seven items for hope, seven items for optimism, nine items for resiliency, and eight items for self-efficacy. After the draft of the scale was completed, five subject matter experts, including four nursing professionals and one psychological professional, were invited to participate. The purpose of the expert review was to test the content validity and construct the validity of the measurement tool and determine if the content adequacy, conceptual clarity, and question meaning were consistent in each subscale [23]. We deferred to their suggestions and reorganised the related item contents accordingly. 

There were 1071 male nursing students studying in 18 nursing schools [24]. We utilised a purposive sampling design to recruit participants from each of the two nursing schools in the Northern, Central, and Southern areas of Taiwan. The follow-up procedure included a quantification administration of the male nursing students we recruited. The data were subjected to item selection by confirmatory factor analysis using LISREL 8.8 software, referring to Li [25], to obtain a formal version of the PsyCap Scale. To understand the status of PsyCap for male nursing students, we collected data from another questionnaire conducted on female nursing students for comparison. Measurement invariance tests were conducted to ensure meaningful comparisons between male and female nursing students on a common scale [26].

### 2.2. Participants

A total of 384 male nursing students from six nursing schools in Taiwan were recruited for this study. They were the main sample participants in the study, with their ages ranging from 18.1 to 23.5 years. The mean age of the sample was 21.00 years, and the standard deviation for age was 0.89. Of the 384 participants, 64.4% (247) had religious beliefs. We also recruited a sample of 402 female nursing students from a nursing school in Taiwan, with their ages ranging from 20.0 to 24.0 years. The mean age of the female nursing students was 20.61 years, with a standard deviation of 1.36. Of the 402 participants, 61.9% (249) had religious beliefs.

### 2.3. Instrument

To develop the PsyCap Scale for male nursing students, we referred to the existing literature [14,27,28,29,30,31] to obtain the connotations of the four subscales. Connotations for hope included a positive motivation state under which one orients to their goal to realize it; when the way to their goal is not available or is thwarted, they find another way, and never give up. Optimism was characterised by positive psychological characteristics that help individuals to cope with stress and adverse conditions. It is also a kind of working theory or style for attributing positive events and their persistence to the self, while attributing adverse events to external and situational factors. Resiliency was defined as the capability to bounce back or recover quickly from disadvantageous situations. It has important implications for promoting ability, such as obtaining positive energy and persistent belief, as well as the ability to overcome challenges, cope with stress, and endure plight to achieve success. Regarding self-efficacy, this was characterised as the belief in oneself that one can succeed when facing challenges and adversity. This belief stems from self-confidence in accomplishing a specific task.

Therefore, we created a 31-item draft encompassing the four dimensions. The draft of the scale was scored using a five-point Likert scale. Point 1 represented strongly disagree, 2 represented disagree, 3 represented no comment, 4 represented agree, and 5 represented strongly agree. Higher scores represented higher hope, optimism, resiliency, and self-efficacy; therefore, the higher the sum of all items, the higher the PsyCap. The content validity index was calculated according to the opinions and evaluations from the five experts, and it was 0.91.

### 2.4. Statistical Analyses

Structural equation modelling (SEM) was applied in the analyses. We randomly selected 60% (230) of the 384 male nursing students to test the four-factor model of the PsyCap Scale for confirmatory factor analysis. The measurement model of SEM was used to test the fit of the 31 items with the four factors. Li’s [25] item selection strategies, mainly based on a modification index involving factor loadings or item error correlations, were considered in the test. In addition, items with factor loadings lower than 0.60 were eliminated. Once the final items were selected, the total male sample was used to confirm the four-factor model. The common model fit indices are listed [32,33], such as χ^2^/*df* (the ratio of chi-square to degrees of freedom) < 5, comparative fit index (CFI) > 0.90, non-normed fit index (NNFI) > 0.90, adjusted goodness of fit index (AGFI) > 0.90, standardised root mean square residual (SRMR) < 0.06, and root mean square error of approximation (RMSEA) < 0.08.

In addition, to ensure that scores between male and female nursing students could be reliably compared, measurement invariance tests were conducted based on nested models. The steps included factor pattern invariance (configural invariance), factor loading invariance (metric invariance), and intercept invariance (scalar invariance) to ensure meaningful comparisons under a common origin and unit of scale. Measurement invariance tests were conducted mainly on the basis of differences in chi-square values (Δχ^2^) and degrees of freedom (Δ*df*) between the nested models. When Δχ^2^ was not significant, it indicated that constraining parameter estimates to be the same between genders was plausible because invariance had been built up. Once the measurement invariances were built up, the mean differences of the four factors could be acquired between genders.

## 3. Results

The four-factor model with 31 items was first tested using confirmatory factor analysis. The model fit indices showed χ^2^ = 868.54, *df* = 428, *p*-value < 0.001, χ^2^/*df* = 2.03, CFI = 0.98, NNFI = 0.98, AGFI = 0.85, SRMR = 0.045, and RMSEA = 0.052, indicating poor model fit. The factor loadings ranged from 0.59 to 0.78, with a mean factor loading of 0.67. The correlation coefficients among the four factors ranged from 0.75 to 0.92. The Cronbach’s alpha reliability coefficients ranged from 0.85 to 0.89 for the four subscales and 0.95 for the total scale. 

To promote the convenient use of the scale and increase the model fit to avoid validity shrinkage in other applied research, fewer and better items would be suitable. In the item selection process, eight items were eliminated because of high correlation between item errors, four items were eliminated because of loadings on non-principal factors, and three items were eliminated because of lower factor loadings. Finally, 16 items were selected based on the modification index of the SEM. The model fit indices for the model of the four factors with 16 items were χ^2^ = 142.20, *df* = 98, *p*-value = 0.02, χ^2^/*df* = 1.45, CFI = 0.99, NNFI = 0.99, AGFI = 0.94, SRMR = 0.035, and RMSEA = 0.034, indicating better model fit outcomes in general. The contents of the 16 items with four factors, their factor loadings, and reliability coefficients are presented in Table 1. It shows that the factor loadings ranged from 0.62 to 0.78, with a mean factor loading of 0.68, indicating better convergent validity. The Cronbach’s alpha reliability coefficients were from 0.75 to 0.83 for the four subscales and 0.91 for the total scale, still indicating good internal consistency reliability. The correlation coefficients among the four factors ranged from 0.73 to 0.91. Although the correlation coefficients were a little high in general, the four-factor model passed the discrimination validity test compared with three-factor models by chi-squared difference tests (not shown in the tables).

Next, to avoid measurement errors interfering in male nursing students’ PsyCap Scale scores for comparison with those of female nursing students, we conducted a series of measurement invariance tests (Table 2). The top panel of Table 2 shows that the models for both male and female nursing students passed tests involving configural invariance, metric invariance, and scalar invariance. Specifically, the good model fit indices of Model A, such as χ^2^ = 351.03, *df* = 196, *p*-value < 0.001, χ^2^/*df* = 1.79, CFI = 0.99, RMSEA = 0.045, and SRMR = 0.035 for male and 0.043 for female nursing students, indicated the same number of factors for both genders (configural invariance). Model B was compared to Model A with Δχ^2^ = 9.74 and Δ*df* = 12, with a non-significant result (*p* > 0.05), indicating the same factor loadings between genders (metric invariance). Additionally, Model C was compared to Model B with Δχ^2^ = 20.24 and Δ*df* = 12, also having a non-significant result (*p* > 0.05), indicating the same intercepts between genders (scalar invariance).

After the three invariance tests involving the A, B, and C models were confirmed, the variances, covariances, and means of the four factors were compared between genders in the common unit (by metric invariance) and origin (by scalar invariance) of factors in the scale. The lower panel in Table 2, also called the structure invariance test, shows comparisons between genders in the variances, covariances, and means of the four factors, specifically when the same factor variances between genders were not met (Δχ^2^ = 21.25, Δ*df* = 4, and *p* < 0.001). The partial factor variance invariance test indicated that two factors, hope and optimism, did not have the same factor variances between genders (Δχ^2^ = 5.56, Δ*df* = 2, and *p* > 0.05). Although the same factor covariances between genders were not met (Δχ^2^ = 19.73, Δ*df* = 6, and *p* < 0.001), the partial factor covariance invariance test showed that the three paired factor covariances (resiliency and self-efficacy, hope and optimism, and hope and self-efficacy) did not have the same factor covariances between genders (Δχ^2^ = 3.32, Δ*df* = 3, and *p* > 0.05). Although the same factor means between genders were not met (Δχ^2^ = 40.85, Δ*df* = 4, and *p* < 0.001), the partial factor mean invariance test showed that the two factors, optimism and resiliency, did not have the same factor means between genders (Δχ^2^ = 4.21, Δ*df* = 2, and *p* > 0.05).

The estimated coefficients in the common metric completely standardised solution are presented in detail in Table 3 and Table 4. Male nursing students had significantly higher optimism (2.27 vs. 2.15) and resiliency (2.28 vs. 2.17) factor means than female nursing students and significantly higher correlation coefficients between hope and optimism (0.92 vs. 0.74) and between hope and self-efficacy (0.89 vs. 0.76). The male nurses also had lower correlation coefficients between resiliency and self-efficacy (0.82 vs. 0.98) than the females.

## 4. Discussion

The PsyCap Scale was developed in this study with good reliability and validity. As shown in Table 4, the high correlation coefficients among the four factors showed that hope, optimism, resiliency, and self-efficacy had much variance overlapping, which reflected a common construct source, namely PsyCap. The study also implied that the definitions of the four constructs (factors) were similar, and they were included under the bigger construct, PsyCap. Nonetheless, although the high correlation coefficients among the four factors seemed to hinder discrimination between them at a glance, they met the test of discrimination validity between nested models with Δχ^2^. In fact, from the perspective of observed variables, the correlation coefficients among the four subscales were only from 0.58 to 0.69 for male nursing students and from 0.56 to 0.64 for female nursing students, and this was consistent with the general extent of correlation coefficients in most subscales summed for a total score for any scale.

The value of the measurement invariance test between genders lies in ensuring comparability of psychological scale scores between male and female nursing students. The invariance test of error that was not met (Model D) only hinders comparisons between individuals but not between groups in observed scores. Although invariances of factor loadings and intercepts between groups were met, they must be sufficient to guarantee between-group differences for factors (latent variables) and observed variables on a common scale because the expected value of measurement errors in group type was zero [33]. In addition, comparisons between genders in the means of latent variables also need to be based on the invariance of variances. Similar to the general t-test of differences in the group means of observed variables, equal variances must be assumed between groups in advance. Therefore, the test of invariance of the latent means was conducted based on Model F rather than Models C or E.

Some reports have found that male nursing students feel isolated and sidelined as they encounter more obstacles. They also report that male nurses experience more role stress and have more negative opinions than female nursing students [34,35,36]. However, no study has compared PsyCap between male and female nursing students. We found that male nursing students had higher PsyCap, including hope, optimism, resiliency, and self-efficacy, than female students on average. Statistically significant differences were observed in optimism and resilience. Hence, traditional occupations and social culture did not lower the PsyCap of male nursing students. It is worth noting that comparisons in latent means yield more reliable and valid outcomes because they are free of measurement errors. The results implied that male nursing students may have more potential for nursing care, despite facing more negative experiences and challenges, than female nursing students. However, there are many psychological variables and different skills related to nursing care that could be important for male nursing students to learn. Nonetheless, our research results should encourage more male nursing students to feel confident to persist in nursing. Nowadays, the shortage of nursing manpower is a challenge worldwide, and encouraging men involved in nursing would help to solve the problem [37]. In Taiwan, the growth rate of male nursing students has increased from 0.41% in 1985 to 11.68% in 2019, and over 1000 male students have majored in nursing in recent years [24]. It is believed that the present research results would benefit the government in developing relevant policies for career planning for male nursing students.

In addition, regardless of the perspectives of PsyCap as trait-like [14,38], having states [39,40,41], and requiring integration [30,42], they all agreed that PsyCap could be fostered and developed. Therefore, the design of different education programmes for male and female nursing students can refer to these results. For example, the results showing that male nursing students had higher correlation coefficients between hope and optimism and between hope and self-efficacy could be applied to training programmes. Interventions can be designed for male nursing students with low optimism or self-efficacy to promote their sense of hope to help increase their optimism and self-efficacy. Although the participants in this study were not sampled randomly, the sample size of male nursing students was large enough to be representative in Taiwan. Finally, the present results came from a non-experimental research design and would be difficult to execute in a causal-effect one; hence, more studies should be conducted to confirm them.

## 5. Conclusions

In this study, we developed the PsyCap Scale, which was shown to have good reliability and validity among male nursing students. In addition, because the scale is brief, including only 16 items, it can save time and is well-suited to the increasing use of online survey administration. Notably, based on advanced statistical methods and measurement invariance tests, we found that male nursing students had higher PsyCap levels of optimism and resiliency than female nursing students, and higher correlation coefficients between hope and optimism and between hope and self-efficacy. However, the male nurses had lower correlation coefficients between resiliency and self-efficacy than the female nurses. The results may be valuable in future research and training for male nursing students. In addition, the development of the PsyCap Scale can help male nursing students understand their potential and standing on hope, optimism, resiliency, and self-efficacy, and consequently, have enough confidence to face the many stressors in the learning stages, thus fostering ambition and devotion to their nursing profession. 

## Figures and Tables

**Table 1 ijerph-19-03620-t001:** Factor loadings and reliability coefficients of the PsyCap Scale (N = 384).

Items	Factor Loading	Cronbach’s Alpha
1. I have the power in my heart to support my learning in nursing.	0.76	
2. I struggle to reach the goal of being a nurse	0.78	
3. I think the nursing profession offers me a more promising future	0.75	
4. I would console myself and persist while facing plight in learning	0.68	0.83
5. I can face uncertainty with a positive attitude towards learning nursing	0.68	
6. I can face various difficulties with a positive attitude towards learning nursing	0.64	
7. I can actively search for solutions to problems in my nursing education	0.65	
8. I can maintain a positive belief in my prospects as a nurse in any situation	0.67	0.75
9. I can learn and grow from errors while facing problems	0.69	
10. I can find relevant persons (e.g., teacher, classmate, and senior schoolmate) to help solve problems	0.63	
11. I can face frustrations bravely during my nursing education	0.63	
12. I can recover rapidly from anger to a normal emotion while learning nursing	0.67	0.75
13. I am enthusiastic about learning nursing to realise my dream	0.69	
14. I dedicate myself to my future and goal	0.69	
15. I am confident to cope with accidents in my nursing education	0.70	
16. I believe that I can solve any problems related to learning nursing	0.62	0.77

Note: Hope includes items 1–4, Optimism includes items 5–8, Resiliency includes items 9–12, and Self-efficacy includes items 13–16. The Cronbach’s alpha coefficient for all items was 0.91. The mean (standard deviation) for the Hope subscale is 14.27 (3.51), for the Optimism subscale is 14.53 (3.48), for the Resiliency subscale is 14.29 (3.52), for the Self-efficacy subscale is 13.54 (3.59), and for the PsyCap Scale is 56.63 (12.13).

**Table 2 ijerph-19-03620-t002:** Tests of measurement invariance in the PsyCap Scale between genders.

Models	Compared Model	χ^2^ (*df*)	RMSEA	CFI	SRMR	Δχ^2^ (Δ*df*)
A. Configural invariance		351.03(196)	0.045	0.986	0.035/0.043	
B. Complete metric invariance	A	360.77(208)	0.043	0.986	0.037/0.049	9.74(12)
C. Complete scalar invariance	B	381.01(220)	0.043	0.985	0.037/0.049	20.24(12)
D. Complete invariance of error variances	C	528.11(236)	0.056	0.975	0.044/0.059	147.10(16) ***
E. Partial invariance of error variances	C	390.80(225)	0.043	0.985	0.038/0.051	9.79(5)
F. Complete invariance of factor variances	C	402.26(224)	0.045	0.984	0.083/0.085	21.25(4) ***
G. Partial invariance of factor variances	C	386.57(222)	0.043	0.985	0.056/0.061	5.56(2)
H. Complete invariance of factor covariances	G	406.30(228)	0.045	0.983	0.066/0.068	19.73(6) **
I. Partial invariance of factor covariances	G	389.89(225)	0.043	0.985	0.055/0.064	3.32(3)
J. Complete invariance of latent means	F	443.11(228)	0.049	0.980	0.082/0.084	40.85(4) ***
K. Partial invariance of latent means	F	406.47(226)	0.045	0.983	0.083/0.085	4.21(2)

** *p* < 0.01. *** *p* < 0.001.

**Table 3 ijerph-19-03620-t003:** Invariant and non-invariant factor loadings, intercepts, error variances, and mean differences between genders.

Factors	Items	Factor Loadings	Intercepts	Error Variances	Latent Mean
Hope	h1	0.68	−0.11	0.46/0.62	
	h2	0.69	−0.24	0.38/0.66	2.49
	h3	0.65	0.13	0.43/0.70	
	h4	0.64	0.23	0.60	
Optimism	o5	0.61	0.05	0.52/0.77	
	o6	0.59	0.21	0.65	2.27/2.15
	o7	0.59	0.01	0.59/0.71	
	o8	0.63	−0.27	0.59	
Resiliency	r9	0.62	−0.13	0.50/0.71	
	r10	0.60	0.14	0.65	
	r11	0.59	0.23	0.65	2.28/2.17
	r12	0.59	−0.21	0.51/0.77	
Self-efficacy	e13	0.61	0.04	0.50/0.76	
	e14	0.64	−0.22	0.48/0.70	2.25
	e15	0.61	0.05	0.50/0.75	
	e16	0.58	0.13	0.58/0.74	

Note: All estimates are presented as a common metric and completely standardised solution. For factor loadings, intercepts, and error variances, non-invariant estimated coefficients are presented with a male/female pattern, and invariant estimated coefficients are presented with a single value.

**Table 4 ijerph-19-03620-t004:** Interfactor correlation of the four factors of the PsyCap scale between genders.

Factors	Hope	Optimism	Resiliency	Self-Efficacy
Hope	1.22/0.79			
Optimism	0.92/0.74	1.13/0.88		
Resiliency	0.78	0.95	1.00	
Self-Efficacy	0.89/0.76	0.91	0.82/0.98	1.00

Note: All estimates are presented in a common metric completely standardised solution; hence, some values are higher than 1.00. Non-invariant estimated coefficients between genders are presented with a male/female pattern; invariant estimated coefficients are presented with a single value. Correlation with hope or optimism should be interpreted cautiously because of the non-invariance of variance in hope and optimism between genders.

## Data Availability

The data presented in this study are available on request due to privacy restrictions.

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
