# Peer review of "Development of the Psychological Capital Scale for Male Nursing Students in Taiwan and Testing Its Measurement Invariance between Genders"

_ijerph, 2022, doi:10.3390/ijerph19063620_

Round 1

Reviewer 1 Report

Review ID Manuscript: ijerph-1623393

Title: Development of the Psychological Capital Scale for Male Nursing Students in Taiwan: Tests of Measurement Invariance Between Genders

Authors present the development of a Psychological Capital Scale for Male Nursing Students in Taiwan and the results of the measurement invariance between male and female nursing students.  The methodological part is the strongest in the manuscript, with some aspect that need to be addressed in the other sections

TITLE

The title, as it is, seems to suggest that the authors present just the measurement invariance between genders. My advice is to change the “colon” by a “full stop” or an “and” as a way to indicate that in the manuscript the readers will find both, the development of a scale and the analysis of invariance between genders. 

ABSTRACT

The abstract presents the more relevant information about the objectives and results of the study. I find it adequate.

INTRODUCTION

The introduction is short but I think it is appropiate for the aim of the study. Some aspects to be modified are the following:

Line 73: “PsyCap” should be written as “Psychological Capital (PsyCap) the first time it is mentioned”

METHODS

The different sub-sections of the Methods section are adequately presented and described.

RESULTS

Some aspects can be improved in this section

Lines 174-175: Authors should mention in the manuscript which items were removed from the final scale and why. A better explanation is needed.

Table 2:

  1. There is a typo in the table: “al-pha” should be “alpha”
  2. The horizontal lines between the items in the same factor should be removed to help readers identify the four items of each factor. Also, include Cronbach Alpha in the last line where the last item of each factor is presented.

Table 3:

  1. Again, there is a typo involving a hyphen. “load-ings” should be “loadings”
  2. Capital letters should be used in the words: items, intercept, hope, optimism, resilience and self-efficacy to make them consistent with the rest of the terms presented in the table.

DISCUSSION

I find that the statement: “The results implied that male nursing students were, in fact, more suitable for nursing care despite facing more negative experiences and challenges than female nursing students”, mentioned in the Discussion, is not entirely justified.

There are many psychological variables and many different skills in the context of care that could be important for nurses. Therefore, the higher scores in hope, optimism, resilience and self-efficacy in male nursing students found in the study are not enough to justify the statement. Authors should, at the very least, rewrite the sentence.

CONCLUSION

Similarly to the Introduction section, I find the Discussion section short but to the point and entirely adequate.

REFERENCES

This section of the manuscript requires extensive work. First, clearly the references in the text of the manuscript follow a different referencing style than is used in the References section. References are numbered in the References section but no numbers are used in the text.

Throughout the text several references including three authors are written in full when the convention is to use “et al, year”: (Bakker, Lyons & Conlon, 2017), Luthans, Youssef, and Avolio (2006) in the Introduction section, and Luthans, Luthans & Chaffin, 2019 in the Discussion section.

A revision of the Reference section is necessary to eliminate typos. Some words that have to be in capital letter are not, period punctuation are not necessary in some places. (Line 327: I nt. Instead of Int.; line 362: Psychology should be in Capital letters because is the name of a journal; line 416, Nurs.e should be Nurse)

The whole manuscript should be reviewed regarding capital letters. They should be used when the name of a scale is written.

Author Response

Dear Reviewer

Reviewer 2 Report

The work is original and performs appropriate analyzes of the data obtained. However, it has some shortcomings, which have to be improved. These shortcomings are as follows: Firstly, methodological characteristics such as the type of research, methodology, method or research design are not sufficiently specified, in addition, no aspect referring to the construction of the categories is mentioned, which give rise to the different items, which, on the other hand, is the fundamental element, when it comes to exposing the construction of the data collection instrument, in addition, the characteristics of the experts and their selection process are not mentioned . On the other hand, the data on the student population, the type of sampling and the characteristics of the student sample are not referenced. Finally, although the discussion has been specified, the conclusions, which is one of the substantial elements of any investigation, is too brief and has little entity. Therefore, it is recommended that each of these elements be improved.

Author Response

Dear Reviewer:

Round 2

Reviewer 2 Report

After the modifications made to the document, there would only be two unresolved issues, since it states that the study used a quantitative research design and, as can be seen, there is a fundamental error in this expression. On the other hand, the definition of the following elements is still lacking: type, methodology, method and research design.

Author Response

Dear Reviewer:

thanks 
